Exploring the potential role of ENPP2 in polycystic ovary syndrome and endometrial cancer through bioinformatic analysis

http://orcid.org/0000-0002-8231-2600 Zhang Xumin
http://orcid.org/0000-0001-8163-7529 Liu Jianrong Liujianrong3@sxmu.edu.cn
Bai Chunmei
Li Yang
Fan Yanxin
The Fifth Clinical Medical College of Shanxi Medical University , TaiYuan, ShanXi , China
Uversky Vladimir
Electronic publication date: 2024 Dec 20
Publication date: 2024
Volume: 12
Electronic Location ID: e18666
Received 2024 Jul 24; Accepted 2024 Nov 18
Copyright: © 2024 Zhang et al.
Copyright year: 2024
Copyright holder: Zhang et al.
License: This is an open access article distributed under the terms of the Creative Commons Attribution License, which permits unrestricted use, distribution, reproduction and adaptation in any medium and for any purpose provided that it is properly attributed. For attribution, the original author(s), title, publication source (PeerJ) and either DOI or URL of the article must be cited.
License URL: https://creativecommons.org/licenses/by/4.0/

Keywords: Ferroptosis, Endometrial cancer, Polycystic ovary syndrome, Bioinformatic analysis, ENPP2

Funding: Shanxi Provincial Central Guidance Local Science and Technology Development Project YDZJSX2022A069 This work was supported by Shanxi Provincial Central Guidance Local Science and Technology Development Project. (Grant No. YDZJSX2022A069). The funders had no role in study design, data collection and analysis, decision to publish, or preparation of the manuscript.

==============================
Background

Growing evidence indicates a significant correlation between polycystic ovary syndrome (PCOS) and endometrial carcinoma (EC); nevertheless, the fundamental molecular mechanisms involved continue to be unclear.

Methods

Initially, differential analysis, the least absolute shrinkage and selection operator (LASSO) regression, and support vector machine-recursive feature elimination (SVM-RFE) algorithms were employed to identify candidate genes associated with ferroptosis in PCOS. Subsequently, the TCGA-UCEC data were utilized to pinpoint the core gene. Then, the expression of ENPP2 in granulosa cells and endometrium of PCOS was validated using real-time PCR (RT-qPCR). Additionally, we investigated the role of ENPP2 in the progression from PCOS to EC through western blotting (WB), colony formation assay, cell scratch assay, transwell assay, and immunofluorescence (IF). Subsequently, ENPP2 gene set enrichment analysis (GSEA) analyses were conducted to identify common pathways involved in PCOS and EC, which were then verified by RT-qPCR. Finally, immune infiltration and the tumor microenvironment (TME) were explored to examine the involvement of ENPP2 in EC progression.

Results

The datasets TCGA-UCEC (pertaining to EC), GSE34526, GSE137684, and GSE6798 (related to PCOS) were procured and subjected to analysis. The gene ENPP2 has been recognized as the shared element connecting PCOS and EC. Next, we observed a significant downregulation of ENPP2 expression in the granulosa cells in PCOS compared to the normal patients, while an upregulation of ENPP2 expression was observed in the endometrium of hyperandrogenic PCOS patients relative to the normal. In vitro, the WB revealed that 5-dihydrotestosterone (DHT) upregulated ENPP2 expression in Ishikawa and HEC-1-A cells. Additionally, we found that ENPP2 promoted the proliferation, migration, and invasion of Ishikawa and HEC-1-A cells. Subsequently, we discovered that overexpressed ENPP2 may lead to an increase in CYP19A1 (aromatase) and AR mRNA level. IF demonstrated that ENPP2 increased the expression of AR, suggesting a regulatory role for ENPP2 in hormonal response within PCOS and EC. Our findings indicated a significant correlation between ENPP2 expression and the modulation of immune responses.

Introduction

Endometrial cancer (EC) represents the most common gynecological tumor encountered among women. Global statistics from 2020 indicate that EC constitutes 4.5% of all cancers in women (Sung et al., 2021). Understanding the underlying pathological mechanisms and treatment options for EC has been a focal point among scholars and experts. Polycystic ovarian syndrome (PCOS) is the most prevalent endocrine and metabolic disorder in women of childbearing age, with an incidence ranging from 6% to 20% (Escobar-Morreale, 2018). Individuals with PCOS face a significant risk of developing endometrial cancer, as women diagnosed with PCOS are three times more likely to develop this condition compared to those without it (Haoula, Salman & Atiomo, 2012). Although breast and ovarian cancers are more common among women than EC, women diagnosed with PCOS have a higher likelihood of developing EC compared to their peers (Barry, Azizia & Hardiman, 2014).

Ferroptosis represents a distinct form of cellular demise that is independent of the apoptotic pathway. This phenomenon can be triggered by cellular stress arising from atypical metabolic and biochemical processes (Wickenheisser et al., 2005). Extensive research has elucidated the pivotal role of ferroptosis in the pathogenesis and progression of cancer and metabolic disorders. Higher levels of iron concentration have been observed in EC tumor tissues compared to non-cancerous tissues (Genkinger, Friberg & Goldbohm, 2012; Xu et al., 2016). The intake of heme iron and total iron content in women are positively correlated with the risk of developing EC. PCOS, the most prevalent endocrine and metabolic disorder in women of childbearing age, may lead to disrupted iron homeostasis and mild iron overload due to menstrual irregularities, insulin resistance, and elevated levels of androgens, which can induce hyperandrogenism and insulin resistance-induced ferroptosis (Escobar-Morreale et al., 2005; Luque-Ramírez et al., 2011; Fernández-Real, López-Bermejo & Ricart, 2002). Importantly, ferroptosis may be implicated in reproductive system dysfunction observed in PCOS. Recent studies indicate that ferroptosis may play a role in the pathological mechanisms associated with normal follicular atresia, which could lead to diminished ovulation or anovulation in individuals with PCOS (Wang et al., 2023). Furthermore, it may also affect the endometrial immune microenvironment and reproductive outcomes in patients diagnosed with PCOS (Zhang et al., 2022). Given the critical role of ferroptosis in the pathogenesis of PCOS and EC, we hypothesized that ferroptosis may be a mechanism linking the two diseases.

Utilizing bioinformatic analysis, we discerned ENPP2 as a gene associated with ferroptosis, which exhibits a significant correlation with both PCOS and EC. ENPP2 is situated on chromosomal region 8q24 and comprises 26 introns and 27 exons (Panagopoulou et al., 2021). ENPP2 encodes autotaxin, an enzyme that catalyzes the synthesis of extracellular lysophosphatidic acid (LPA), which is implicated in a diverse range of biological activities associated with tumorigenesis and progression, including enhanced cellular survival, angiogenesis, lipid metabolism, glucose metabolism, mitochondrial functionality, invasion potential, and metastatic capabilities (Kaffe, Magkrioti & Aidinis, 2019). Since ATX was first discovered as a cell motility factor, it has been found to be expressed by a wide variety of tumor cells, including those in breast cancer, glioblastoma, hepatocellular carcinoma, neuroblastoma, non-small-cell lung cancer, prostate cancer, renal cell carcinoma and thyroid carcinoma (Yuelling & Fuss, 2008). At the epigenetic level, the increased methylation of ENPP2 may significantly contribute to unfavorable prognostic indicators in cancer and pathogenesis (Panagopoulou et al., 2021). Therefore, we collected ovarian granulosa cells and endometrial tissue samples from non-PCOS and PCOS patients to investigate the differential expression of ENPP2 between the two groups. Moreover, in vitro cellular investigations were undertaken to clarify the fundamental mechanisms linked to the transition from PCOS to EC.

Materials and Methods

Data collection

We initially acquired PCOS datasets from the GEO database (http://www.ncbi.nlm.nih.gov/geo), selecting three specific datasets: GSE34526, GSE137684, and GSE6798. The R packages “sva” and “limma” were employed to integrate and standardize the two datasets. We obtained 259 ferroptosis-related genes (FRGs) from the FerrDb database (Zhou & Bao, 2020), which included genes associated with driver genes and suppressor genes. Differentially expressed genes (DEGs) were identified using the R package “limma”. Subsequently, the candidate genes (CGs) were obtained by intersecting FRGs with DEGs for further investigation. Additionally, we extracted uterine corpus endometrial carcinoma (UCEC) data from the cancer genome atlas program (TCGA) cohort (Tomczak, Czerwińska & Wiznerowicz, 2015), which includes 589 cases of endometrial cancer and serves as a comprehensive database for cancer gene information and expression data. A flowchart outlining our study design was presented in Fig. 1.

Figure 1 Flow chart of this study.

Abbreviations: PCOS, polycystic ovary syndrome; GEO, Gene Expression Omnibus; LASSO, least absolute shrinkage and selection operator; SVM-RFE, support vector machine–recursive feature elimination; TCGA, the cancer genome atlas; UCEC, uterine corpus endometrial carcinoma; CGs, candidate genes; FRGs, ferroptosis-related genes; DEGs, differentially expressed genes; TME, tumor microenvironment.

Identification of the core gene

The least absolute shrinkage and selection operator (LASSO) and the support vector machine–recursive feature elimination (SVM-RFE) algorithms, executed through the ‘glmnet’ and ‘e1071’ packages in R, were utilized to discern significant candidate genes (CGs) pertinent to the diagnosis of PCOS (Chen et al., 2022). To facilitate the clinical application, a nomogram model was developed using the “root-mean-square” package. Additionally, decision curve analysis (DCA) and clinical impact curve (CIC) were employed to assess both modeling stability and value.

Subsequently, we analyzed the expression of three key CGs in TCGA-UCEC for EC and in the GSE34526 and GSE137684 datasets for PCOS to identify the core gene between PCOS and EC. The diagnostic performance of these key CGs was evaluated based on their area under the receiver operating characteristic curve (AUC), and receiver operating characteristic (ROC) curves were plotted to illustrate their discriminative ability.

Gene set enrichment analysis

We performed gene set enrichment analysis (GSEA) utilizing the “clusterProfiler” package to discern the shared pathways implicated in both diseases. GSEA was performed using c2.cp.kegg.symbols.gmt to obtain the signaling pathways with both diseases. Enrichplot was employed to show the top five activating and inhibiting pathways associated with the core gene in the two diseases.

The relationship between the ENPP2 and immune infiltration

We utilized TIMER2.0 (http://timer.cistrome.org/) (Li et al., 2017) and TISIDB (http://cis.hku.hk/TISIDB/) (Ru et al., 2019) to explore the association between ENPP2 and immune cell infiltration, immunostimulators, immunoinhibitors, and chemokines (or receptors) in EC. Additionally, we employed the “estimate” R package to analyze the correlation between gene expression levels and the tumor microenvironment.

Specimen collection

This research encompassed eight individuals with normoandrogen (NA) and hyperandrogen (HA) PCOS, alongside four women with normal ovulatory function, all of whom participated in in vitro fertilization (IVF) between August 5th, 2023, and October 5th, 2023, at the Fifth Clinical Medical College of Shanxi Medical University. PCOS is diagnosed based on the Rotterdam 2003 diagnostic criteria (The Rotterdam ESHRE/ASRM-Sponsored PCOS Consensus Workshop Group, 2004). All patients received human chorionic gonadotropin (HCG) 36 h prior to vaginal ultrasound-guided follicular fluid aspiration. Granulosa cells were isolated from the follicular fluid. The endometrium was obtained by curettage at the proliferative stage of the menstrual cycle. All experimental protocols were approved by our ethical organization namely the Ethics Committee of Fifth Hospital of Shanxi Medical University (no. 2023288V1.0). Written informed consent was obtained from all participants.

Cell culture and treatments

The HEC-1-A and Ishikawa (ISK) cell lines, obtained from ATCC (Rockville, MD, USA), were cultivated in a humidified incubator with 5% CO2 at 37 °C. Following subculturing, cells in the logarithmic phase were utilized for the subsequent experiments.

HEC-1-A and ISK cells were treated with 100 nM 5-dihydrotestosterone (DHT) (MCE, Shanghai, China) based on a previously reported concentration to induce AR receptor-mediated effects in human endometrial tissue in vitro (López-Janeiro, 2021), or 10 nM ONO-8430506 (MCE, Shanghai, China) which is an ATX/ENPP2 inhibitor.

Cell transfection

Nonspecific control (NC) and small interfering RNAs (siRNAs) directed against AR were synthesized by GenePharma (Shanghai, China). The ENPP2 overexpression plasmid (OE-ENPP2) and the overexpression normal control (OE-NC) were acquired from GenePharm (Shanghai, China). For transient cell transfection, HEC-1-A and ISK cells were seeded in 6-well plates, and were randomly divided into the following four groups: OE-NC group, transfected with OE-NC construct; OE-ENPP2 group, transfected with OE-ENPP2 construct; si-NC group, transfected with si-NC construct; si-AR group, transfected with si-AR construct. After 24 h of incubation. Transfection was performed using Lipofectamine 3,000 (Invitrogen, Carlsbad, CA, USA). To establish stably infected cell lines with OE-ENPP2, OE-ENPP2 lentivirus were infected into HEC-1-A and ISK cells with 4 μg/mL polybrene assistance and selected with 2.5 μg/mL puromycin. All sequence information was provided in Table S1.

Real-time PCR analysis

The TRIzol Reagent (APPLYGEN, Beijing, China) facilitated the extraction of total RNA from cells and tissues, subsequently undergoing isopropanol precipitation. The isolated RNA levels were measured using ND2000 (Thermo, Waltham, MA, USA), and then 1 μg of total RNA was reverse-transcribed into complementary DNA using the FSQ-301 reverse transcription kit (TOYOBO, Japan). SYBR Green premix Ex Taq II (Takara Bio Inc., Otsu, Shiga, Japan) was added to each reverse transcription product, and the reaction mixture was then amplified using a CFX96TM Real-Time PCR detection system (Bio-Rad, Hercules, CA, USA). The relative gene expression was determined by calculating 2−ΔΔCt. The following primer pairs were used for amplification:

ENPP2 Forward: 5′-CCTGCAGTGCTTTATCGGAC-3′

Reverse: 5′-GAGAAACACGGACATCAGGC-3′

AR Forward: 5′-GGTGAGCAGAGTGCCCTATC-3′

Reverse: 5′-GCAGTCTCCAAACGCATGTC-3′

CYP19A1 Forward:5′-AGACGCAGGATTTCCACAGA-3′

Reverse: 5′-GGTCACCACGTTTCTCTGCT-3′

beta-actin Forward:5′-TCTCCCAAGTCCACACAGG-3′

Reverse: 5′-GGCACGAAGGCTCATCA-3′

Repetitive amplifications were conducted on serially diluted amplicons to create standard curves that enable the quantification of PCR products. The ENPP2, CYP19A1, and AR mRNA expression levels were normalized to beta-actin values. The real-time PCR conditions were as follows: 40 cycles of 95 °C for 30 min, 95 °C for 5 s, 60 °C for 30 s, and 72 °C for 30 s.

Western blot assay

The granulosa cells and endometrial tissue from patients and cell lines underwent disruption using RIPA lysis buffer (AR0102-10, BOSTER, China) enriched with a protease inhibitor cocktail (SigmaAldrich; Merck KGaA, Germany), followed by centrifugation at 12,000 ×rpm at 4 °C for a duration of 15 min. The protein bands were transferred onto polyvinylidene difluoride membranes after separating equal amounts of total proteins using 10% sodium dodecyl sulfate-polyacrylamide gel electrophoresis (IPVH00010; Merck Millipore, Burlington, MA, USA). Target bands were incubated with corresponding primary antibodies against ENPP2 (14,243-1-AP, 1:1,000, Proteintech, China) overnight at 4 °C, followed by the addition of HRP-labeled secondary antibodies. The protein bands were visualized using an ChemiDocTM XRS+ Imaging System (BIO-RAD, Hercules, CA, USA).

Colony formation assay

Cells in the logarithmic growth phase were seeded in six-well culture plates. The culture medium was changed every 3 days and the cells were cultured for 10 days or until more than 50 monoclonal cells were obtained (37 °C, 5% CO2). Subsequently, the cells were fixed with 4% paraformaldehyde and stained with 0.1% crystal violet. The rate of cell proliferation was assessed by the ImageJ software.

Cell scratch assay

A total of 1 × 106 HEC-1-A and ISK cells were seeded in six-well plates, and scratches were created using a sterile pipette’s yellow tip when the cells reached 90% confluence. After incubation for 12 to 24 h in medium containing 10% FBS, images of the same location were captured using a microscope to assess wound closure. The rate of cell migration was calculated by analyzing the healing area with ImageJ software.

Transwell

The HEC-1-A and ISK cells were reconstituted in a serum-free medium, and a suspension containing 5 × 10 ^ 4 cells was introduced into the upper chamber of a 24-well plate. The lower chamber was filled with a medium comprising 10% serum, while the upper chamber featured a membrane with an 8 μm pore size, coated with diluted Matrigel (Millipore, BD, USA). After incubation for 24 h, the cells were fixed with 4% paraformaldehyde for 20 min and stained with 0.5% crystal violet for another 20 min. Five random fields of view were selected from each chamber for cell counting.

Immunofluorescence

The anti-AR immunofluorescence assays were performed in accordance with the manufacturer’s guidelines (Abcam, 1:250). The principal antibody utilized in this study was directed against AR (ab18394). Subsequently, the cells were exposed to corresponding FITC-conjugated secondary antibodies. After a 2-h incubation period, the nucleus was stained with 0.1% DAPI for 30 min. Confocal microscopy (OLYMPUS, Tokyo, Japan) was utilized to capture images.

Statistical analysis

The statistical analyses were executed utilizing R software (version 4.2.2). Each experiment was conducted at least three times. GraphPad Prism 9.00 (GraphPad Software Inc., San Diego, CA, USA) was utilized for the evaluation of the statistical data. The student’s t-test or one-way analysis of variance (ANOVA) were used to detect significant differences between two independent groups. All P values were calculated as two-sided, and a significance level of p < 0.05 was considered statistically significant.

Results

Differentially expressed ferroptosis-related genes in PCOS

Initially, the normalization of the GSE34526 and GSE137684 datasets was conducted, as illustrated in Fig. S1. Subsequently, we employed the normalized data to examine the disparities in gene expression between PCOS and normal samples, corroborating the findings through validation with GSE6798. The cutoff criteria were defined as follows: false discovery rate (FDR) <0.05 and log2 |FC| > 1. Using the “limma” package, we identified 741 DEGs in women with PCOS compared to those without, which were further visualized using volcano plots and heat maps (Figs. 2A and 2B). We then cross-referenced these DEGs with a panel of 259 ferroptosis-related genes from FerrDbV1 database, resulting in the identification of nine common genes (CAPG CYBB, ENPP2, HAMP, HMOX1, LOC390705, NCF2, PLIN4, TLR4). A Venn diagram was used to illustrate this intersection (Fig. 3A). Finally, we generated a boxplot to compare the levels of expression for these nine genes in both normal and PCOS patients (Fig. 3B). We observed significant upregulation of seven CGs and downregulation of two CGs in PCOS samples compared to the normal control group.

Figure 2 Volcano plot and heatmap of DEGs.

(A)The volcano plot shows the DEGs in integrated dataset (the red dots represent the upregulated genes, and the blue dots represent the downregulated genes). (B) Supervised hierarchical clustering for 741 unique probe sets was performed by Ward’s method and Euclidean Distance as the distance measure. Each row represents a single gene; each column represents a sample. Expression values were color coded: blue, transcript level below the median; and red, greater than median.

Figure 3 View of nine CGs in PCOS.

(A) The intersection of DEGs in PCOS and ferroptosis related genes. A Venn diagram was constructed using VennDiagram package. (B) Boxplot of the mRNA levels of nine CGs in PCOS samples. Boxplots are determined by the spacing between quarterbacks, with the median line representing the median and the whisker being 1.5 times the quarterback spacing. T test for difference significance. *p < 0.05, **p < 0.01, ***p < 0.001.

Diagnostic biomarkers of PCOS

Two distinct algorithms, LASSO logistic regression and SVM-RFE, were utilized to assess nine candidate genes linked to PCOS. The application of the LASSO logistic regression algorithm led to the identification of a quartet of genes that may serve as potential biomarkers (Fig. 4A). Similarly, the SVM-RFE algorithm also selected four genes as effective CGs (Fig. 4B). Three overlapping CGs (HAMP, ENPP2, HMOX1) were subsequently identified as promising diagnostic biomarkers for further research (Fig. 4C). These overlapping genes screened by both SVM-RFE and LASSO demonstrated reliability in predicting PCOS.

Figure 4 Diagnostic biomarkers of PCOS.

(A) Optimal lambda value was selected in the LASSO regression model based on 10-fold cross-validation. (B) Line graph shows the cross-validated accuracy based on different numbers of CGs in the SVM-RFE model. (C) The overlap genes of LASSO and SVM-RFE algorithms.

To enhance the clinical identification of PCOS through the application of specific genetic markers (HAMP, ENPP2, HMOX1), we devised a nomogram model grounded in the identified CGs (Fig. 5A). The calibration curves of the nomogram model demonstrated excellent agreement between observed and predicted diagnostic rates, indicating the high accuracy of our model for PCOS diagnosis (Fig. 5B). The DCA and CIC indicated that our nomogram model had substantial value in PCOS diagnosis (Figs. 5C, 5D). Subsequently, we validated the differential expression patterns of these three key CGs in GSE6798 dataset (Fig. 6).

Figure 5 Construction of the nomogram model.

(A) Construction of the nomogram model based on HAMP, ENPP2, and HMOX1. (B) Calibration curve showed the accuracy of the nomogram model in the diagnosis of PCOS. (C) Decision curve analysis (DCA) indicated that the nomogram model had better clinical application value in the diagnosis of PCOS. (D) Clinical impact curve (CIC) suggested that the nomogram model had higher clinical application value in the diagnosis of PCOS.

Figure 6 Validating HAMP, ENPP2, and HMOX1 differential expression in the GSE6798 dataset.

Boxplots are determined by the spacing between quarterbacks, with the median line representing the median and the whisker being 1.5 times the quarterback spacing. T test for difference significance. ***p < 0.001.

ENPP2 is the core gene

To elucidate the fundamental gene linking PCOS and EC, we investigated the significance of three pivotal CGs associated with PCOS within the TCGA-UCEC database. After performing gene differential expression analysis, we found a significant difference in ENPP2 (p < 0.05) between EC patients and normal individuals, with high expression levels observed in EC patients. However, there were no significant differences between HMOX1 and HAMP (p > 0.05) (Fig. 7A). The specificity and sensitivity of ENPP2 for diagnosing two diseases were tested by ROC. For PCOS, the AUC values were 0.942 for HAMP, 0.942 for ENPP2, and 0.962 for HMOX1, respectively. For EC, the AUC values were 0.509 for HAMP, 0.594 for HMOX1 (Figs. S2A, S2B), and 0.886 for ENPP2, respectively (Fig. 7B). Furthermore, we validated the expression levels of ENPP2 in our clinical specimens through the utilization of qRT-PCR assays and WB analysis. The findings revealed a notable decrease in ENPP2 expression within the granulosa cells of individuals diagnosed with PCOS when contrasted with the normal cohort (Fig. 7C). Conversely, an increase in ENPP2 expression was noted in the endometrium of HA-PCOS patients in comparison to the normal group (Fig. 7D). Therefore, we considered that androgens might regulate the expression of ENPP2 in Endometrial tissue of PCOS.

Figure 7 Identification of the core gene.

(A) Different expression of HAMP, ENPP2, and HMOX1 in TCGA-UCEC cohort. Boxplots are determined by the spacing between quarterbacks, with the median line representing the median and the whisker being 1.5 times the quarterback spacing. T test for difference significance. (B) Receiver operating characteristic (ROC) curves were used to verify the accuracy of three CGs predictions (HAMP, ENPP2, HMOX1) in PCOS and TCGA-UCEC cohort. (C) qRT-PCR and western blot validated ENPP2 expression on granulosa cells in normoandrogen (NA)/hyperandrogen (HA) PCOS and normal women. (D) qRT-PCR and western blot validated ENPP2 expression on endometrium in NA/HA PCOS and normal women. *p < 0.05, **p < 0.01, ***p < 0.001, ****p < 0.0001.

The role of ENPP2 in EC and PCOS

The transient transfection of small interfering RNA resulted in a reduction of AR expression in ISK and HEC-1-A cells (Fig. 8A). We noted a 100 nM DHT-induced upregulation of ENPP2 expression in HEC-1-A and ISK cells, which was later mitigated by AR knockdown (Figs. 8B, 8C). These findings suggested that the expression of ENPP2 was upregulated in endometrial epithelial cells upon induction by DHT. Stable transfection lines were established by transfecting the OE-ENPP2 into Ishikawa and HEC-1-A cells, respectively (Fig. 9A). Subsequently, stable cell lines OE-ENPP2 and ATX/ENPP2 inhibitor-ONO-8430506 were used for subsequent cell function experiments. ONO-8430506 used a previously reported concentration to efficiently inhibit of lysophosphatidic acid (LPA) formation, with IC50s of approximately 10 nM in vitro (Saga et al., 2014). Colony formation assay demonstrated that OE-ENPP2 significantly enhanced the proliferation of ISK and HEC-1-A cells (p < 0.01), and could be rescued after administration of 10 nM ONO-8430506 (Fig. 9B and Fig. S3A). The results of the scratch assay indicated that OE-ENPP2 markedly promoted ISK and HEC-1-A cells migration (p < 0.001), and could be rescued after administration of 10 nM ONO-8430506 (Fig. 9C and Fig. S3B). Transwell assay demonstrated a significant enhancement in invasion in ISK and HEC-1-A cells (p < 0.01), and could be rescued after administration of 10 nM ONO-8430506 (Fig. 9D and Fig. S3C).

Figure 8 DHT regulates the expression of ENPP2 in ISK and HEC-1-A cells.

(A) The transfection efficiency of AR decreased in mRNA level by qRT-PCR assay (B) 100 nM DHT-induced Up-regulation the expression of ENPP2 in ISK cells. (C) 100 nM DHT-induced Up-regulation the expression of ENPP2 in HEC-1-A cells. **p < 0.01, ***p < 0.001, ****p < 0.0001.

Figure 9 Effect of ENPP2 on ISK cells proliferation, invasion and migration.

(A) The mRNA levels in Overexpression normal controls (OE-NC) and ENPP2 overexpression (OE-ENPP2) ISK cells. (B) OE-ENPP2 ISK cells were continuously cultured for 10 days or until more than 50 monoclonal cells had grown, and 10 nM ONO-8430506 was added to inhibit ENPP2 expression. (C) Migration rates of ISK cells at 12 and 24 h. (D) Transwell invasion rates of ISK cells at 24 h. *p < 0.05, **p < 0.01, ***p < 0.001.

To enhance our understanding of the function of ENPP2 in both conditions, we conducted GSEA analysis for ENPP2 within the PCOS and TCGA-UCEC datasets, respectively. The top 5 up-regulated and down-regulated pathways were visualized in Figs. 10A, 10B, which showed that ENPP2 was involved in retinol metabolism, complement and coagulation cascades, as well as neuroactive ligands interacting with receptors in both disease groups. Previous studies have demonstrated that retinol metabolism can impact the expression and activity of aromatase, which in turn affects the level of androgen (Ciolino, Dai & Nair, 2011; Miyazaki et al., 2016). On the other hand, androgen may affect the levels of retinol and the expression of retinyl-related proteins (Ciolino, Dai & Nair, 2011; Miyazaki et al., 2016; Philips et al., 2015). The metabolic pathway of retinol, which involves ENPP2, can elevate the expression or activity of aromatase, enhancing AR expression. To approve this hypothesis, we performed RT-qPCR experiments on OE-ENPP2 ISK cells and found that compared to the OE-NC group, the expression of AR and CYP19A1 (aromatase) was higher in the OE-ENPP2 group. However, this effect was rescued by 10 nM ONO-8430506 (Fig. 10C).

Figure 10 ENPP2 expression and function.

(A, B) ENPP2 single-gene GSEA in PCOS and TCGA-UCEC. (C) AR and CYP19A1 mRNA levels in OE-ENPP2 ISK cells before and after treatment with 10 nM ONO-8430506. (D) ENPP2 and AR correlation analysis in GEPIA database. (D) The effect of ENPP2 on AR was assessed by immunofluorescence. *p < 0.05, **p < 0.01, ***p < 0.001.

The GEPIA platform, as detailed by Li et al. (2021), demonstrated a positive correlation between the expression levels of ENPP2 and AR in endometrial carcinoma (Fig. 10D). Additionally, we conducted an OE-ENPP2 in ISK cell lines (Fig. 10E) and validated the influence of ENPP2 on AR expression through immunofluorescence. Our results supported a positive correlation between the expression levels of ENPP2 and AR. In a previous phase II study found that the AR inhibitor enzalutamide combined with paclitaxel and carboplatin could significantly alleviate previously untreated advanced or recurrent endometrioid lesions in patients (Westin et al., 2021). Consequently, AR exhibits a notable correlation with tumor aggressiveness and the prognosis of endometrial cancer, positioning it as a promising candidate for understanding tumor progression (Duska, Zadeh & Mills, 2018; Hussain et al., 2018). However, excess androgens are known to cause PCOS pathogenesis by hyperactivating AR (The Rotterdam ESHRE/ASRM-Sponsored PCOS Consensus Workshop Group, 2004). These findings suggested that the ENPP2/AR axis may play a dual role in EC or PCOS progression.

Characteristics of the immune infiltration and tumor microenvironment (TME) in different expression levels of ENPP2

Cancer represents a multifaceted ecosystem, comprising tumor cells alongside diverse non-cancerous cells, all intricately interwoven within an aberrant extracellular matrix that collectively constitutes the tumor microenvironment. Extensive evidence has consistently demonstrated the pivotal role of TME in cancer pathogenesis (Visser & Joyce, 2023). The analysis of TME scores for ENPP2 demonstrated significant differences among varying levels of ENPP2 expression. Notably, EC samples exhibiting high ENPP2 expression displayed significantly elevated TME scores compared to those from normal patients (Fig. 11). Through TIMER2.0, we discovered that ENPP2 was associated with the degree of infiltration of Endothelial (Rho = 0.529, p = 1.16e−07), T cell regulatory (Rho = 0.373, p = 3.45e−04), cancer correlated fibroblast cell (Rho = 0.359, p = 2.57e−10), Hematopoietic stem cell_XCELL (Rho = 0.348, p = 8.98e−04), mast cell activated (Rho = 0.355, p = 6.95e−04) (Figs. 12A, 12B). A correlation was then identified between ENPP2 expression and various immunological signatures in the TISIDB. The expression of ENPP2 in EC demonstrated a significant correlation with various chemokines and their receptors, notably CXCL12 (Rho = 0.469), CCL14 (Rho = 0.515), CXCL21 (Rho = 0.32), and CCL23 (Rho = 0.291) (Figs. 12C–12F). In our investigation, we observed a correlation between ENPP2 expression and various immunostimulators and immunoinhibitors in EC, specifically KDR (Rho = 0.51), CD28 (Rho = 0.307), CD40LG (Rho = 0.302), and CXCL12 (Rho = 0.469) (Figs. 11G–11J). Therefore, it has been confirmed that ENPP2 may play a role in modulating tumor associated immune infiltration in EC.

Figure 11 The correlation analysis between TME score and ENPP2.

*p < 0.05, ***p < 0.001.

Figure 12 Association between ENPP2 and immune infiltration expression in TCGA-UCEC cohort.

(A, B) Correlation of ENPP2 expression with infiltration levels of Endothelial, T cell regulatory, Cancer associated fibroblast cell, Hematopoietic stem cell_XCELL, and Mast cell activated in EC available at TIMER2.0 database. (C–F) Correlation between ENPP2 expression and chemokines and chemokines receptors in EC available at TISIDB database. (G–J) Correlation between ENPP2 expression and immunostimulatory and immunoinhibitory in EC available at TISIDB database.

Discussion

Obesity, nulliparity, age >50, infertility, hypertension, diabetes, chronic anovulation, and unopposed estrogen supplementation are all significant risk factors for the development of EC in women with PCOS. The prevalence of EC in women diagnosed with PCOS varies between 20% and 37%. However, the precise molecular mechanisms contributing to this increased vulnerability remain inadequately elucidated (Navaratnarajah, Pillay & Hardiman, 2008; Papaioannou & Tzafettas, 2010). A research study discovered that the endometrium of women with PCOS and EC showed a significant increase in the expression of IGF1, IGFBP1, and PTEN genes compared to controls, regardless of factors such as body mass index, waist-hip ratio, or systemic measurements of insulin resistance using HOMA-IR (Shafiee, Seedhouse & Mongan, 2016). An additional research study found a significant increase in SREBP1 gene expression in the endometrium of women with PCOS and EC, compared to control subjects.

Ferroptosis is deeply connected to numerous biological processes, particularly abnormal metabolic pathways, and cellular oxidation-reduction mechanisms. Research confirms that ferroptosis effectively impedes the growth of cancer cells, particularly those exhibiting resistance to conventional therapeutic approaches (Stockwell, 2022). Ferroptosis, characterized by dysregulated iron homeostasis, lipid peroxidation, and depletion of antioxidants, plays a pivotal role in the pathogenesis of EC. Prior investigations have established the occurrence of ferroptosis in ECC (López-Janeiro, 2021), with around 80% of patients receiving an initial diagnosis of low-grade endometrioid endometrial carcinomas (Stockwell, 2022; Sung et al., 2021). PCOS is a significant risk factor for EC, and emerging evidence suggests the involvement of ferroptosis in the pathophysiology of PCOS (Wang et al., 2023; Zhang et al., 2022). Understanding the underlying pathophysiological association between PCOS and EC could provide valuable insights for developing clinical treatment strategies for both conditions. Overall, ferroptosis may play a crucial role in the pathogenesis of PCOS and EC.

In this study, we identified nine DGEs associated with ferroptosis in PCOS using GEO datasets. To identify the most suitable model genes, we employed LASSO regression alongside SVM-RFE algorithms. Subsequently, a nomogram model for PCOS was developed based on three specific genes related to ferroptosis: HAMP, ENPP2, and HMOX1; Furthermore, utilizing the TCGA-UCEC dataset, we identified ENPP2 as the core gene.

The ENPP2 gene is responsible for encoding a variety of enzymes, notably Autotaxin (ATX), which is also referred to as oligonucleotide phosphodiesterase/pyrophosphatase family member 2, alongside ENPP3. Both enzymes play a crucial role as phosphodiester bond cleavers and phospholipases (Kawagoe et al., 1995). Many studies have shown that overexpression of ENPP2 inhibits the ability of ferroptosis to prevent the development of hypoxia/reoxygenation (H/R) injury (Fang, Shen & Liao, 2023) and can enhance tumor cell spreading, migration, and metastasis through LPAR1 (Auciello et al., 2019; Lin et al., 2019). ENPP2 is integral to the synthesis of LPA, which has significant implications for reproductive function, particularly in relation to the MAPK/p38 and NF-κB signaling pathways (Chen et al., 2008). It serves a pivotal function in the processes of embryo implantation and the initial stages of pregnancy. We found that ENPP2 mRNA and protein expression were significantly increased in EC/PCOS compared to endometrial tissue from women without PCOS. We cultured HEC-1-A and ISK cells in vitro at a concentration of 100 nM DHT and demonstrated that DHT-induced up-regulation significantly increased the expression of ENPP2. Furthermore, in vitro investigations were performed to explore the functional role of ENPP2 and its ability to enhance tumor proliferation, migration, and invasion were discovered. Taken together, these results indicated that ENPP2 might represent a significant factor in EC and PCOS progression. To further explore the potential role of ENPP2 in PCOS and EC, we performed GSEA analysis. The findings revealed a significant enrichment of ENPP2 in retinol metabolism in both pathological conditions. Previous studies have shown that aberrant retinol metabolism could influence androgen synthesis in theca cells and endometrial differentiation, and there is a reciprocal influence between androgens and retinol (Carter et al., 1996; Ciolino, Dai & Nair, 2011; Miyazaki et al., 2016; Wickenheisser et al., 2005). The results of our investigation revealed that the mRNA levels of AR and CYP19A1 (aromatase) were increased in the OE-ENPP2 ISK cells. Consequently, our research suggests that the metabolic pathway of retinol, which involves ENPP2, may play a role in elevating aromatase expression or activity. This, in turn, could lead to an augmented synthesis of AR, resulting in heightened AR expression. This mechanism warrants further investigation. Many studies have shown that ENPP2 expression level is closely related to hormone response (Liszewska et al., 2009; Seo et al., 2012). Hussain et al. (2018) found that the epithelial cells of the endometrium ENPP2 may be regulated by E2 and play a role in reproductive functions. Interestingly, many studies have demonstrated a strong correlation between AR and the incidence, high-grade lesions, and prognosis of EC (Duska, Zadeh & Mills, 2018; Hussain et al., 2018). The findings from our investigation indicated that the mRNA levels of AR and CYP19A1 (aromatase) were elevated in the OE-ENPP2 ISK cells. As a result, our investigation indicates that the metabolic pathway associated with retinol, particularly involving ENPP2, could be significant in enhancing the expression or activity of aromatase. This, consequently, may facilitate an enhanced synthesis of AR, culminating in an elevated expression of AR.

The mounting evidence suggests that the infiltration of immune cells into tumors plays a pivotal role in driving cancer progression. The study conducted by Panagopoulou et al. (2021) has demonstrated the crucial role of methylation in the regulation of ATX expression in cancer, while also establishing a correlation between ENPP2 expression and the extent of CD8+ T cell infiltration (Garaud et al., 2019; Matas-Rico et al., 2021). Earlier research has shown that the significant involvement of T cell regulatory and cancer-associated fibroblasts in EC (Kolben et al., 2022; Yu et al., 2022). Nonetheless, the role of ENPP2 in immune infiltration within EC remains ambiguous. Consequently, we undertook additional inquiries to explore the relationship between ENPP2 expression and the infiltration of immune cells in EC. According to our findings, the high expression of ENPP2 was found to be crucial for immune regulation in EC. Significantly, a correlation was observed between the expression of ENPP2 and several immune cell types, including endothelial cells, regulatory T cells, cancer-associated fibroblasts, hematopoietic stem cell_XCELLs, and activated mast cells. In conclusion, these findings strongly suggest that ENPP2 may exert a considerable influence on the occurrence, development, and progression of PCOS and EC by regulating ferroptosis, indicating its potential as a novel risk significant gene.

Conclusions

This study presents compelling evidence that ENPP2 serves as a crucial factor and therapeutic target implicated in both PCOS and EC. In vitro, DHT promotes the upregulation of ENPP2 expression, which in turn facilitates the proliferation, migration, and invasion of HEC-1-A and ISK cells. Furthermore, ENPP2 represents a significant gene involved in EC and PCOS progression by regulating the hormonal response, and ENPP2/AR axis may play a dual role in EC or PCOS progression. Additionally, ENPP2 may regulate the tumor immune microenvironment through its interaction with immune cells in EC. While, in this study, as the FRGs screened together, HAMP and HMOX1 were not further discussed. In future research, identifying these genes as combinatorial biomarkers in addition to ENPP2 might improve clinical fidelity and strengthen our conclusion. In this study, we found that AR signaling and ENPP2 have significant potential for clinical application. Future investigations will concentrate on the application of ARSI treatment in ISK cells and animal models of endometrial cancer, aiming to delve deeper into the possibilities for novel therapeutic approaches for this malignancy.

Supplemental Information

Supplemental Information 1 RT-qPCR_data and MIQE checklist.

Supplemental Information 2 AR-siRNA and pcDNA3.1-ENPP2 Primer sequences.

Primer sequences of ENPP2 for reverse transcription-quantitative polymerase chain reaction. AR-siRNA and pcDNA3.1-ENPP2 Primer sequences.

Supplemental Information 3 Normalized for the datasets GSE34526, GSE137648, and GSE6798.

Supplemental Information 4 Verified the accuracy of HAMP and HMOX1 in TCGA-UCEC cohort by receiver operating characteristic (ROC) curves.

Verified the accuracy of HAMP and HMOX1 in TCGA-UCEC cohort by receiver operating characteristic (ROC) curves.

Supplemental Information 5 Effect of ENPP2 on HEC-1-A cells proliferation, invasion and migration.

(a) The mRNA levels in overexpression normal control (OE-NC) and ENPP2 overexpression (OE-ENPP2) HEC-1-A cells. (b) OE-ENPP2 HEC-1-A cells were cultured for 10 days or until more than 50 monoclonal cells, and 10nM ONO-8430506 was added to inhibit ENPP2 expression. (c) Migration rates of HEC-1-A cells at 12 h and 24 h. (d) Transwell invasion rates of HEC-1-A cells at 24 h. *p<0.05, **p<0.01, ***p<0.001

Supplemental Information 6 Raw figure for western blot.

Supplemental Information 7 Raw data of each experimental result.

We wish to convey our gratitude to all faculty members who have extended their support and assistance. We extend our gratitude for the invaluable support provided by the proofreaders and editors.

Additional Information and Declarations

Competing Interests

Author Contributions

Human Ethics

Data Availability

The authors declare that they have no competing interests.

Xumin Zhang conceived and designed the experiments, performed the experiments, analyzed the data, prepared figures and/or tables, authored or reviewed drafts of the article, and approved the final draft.

Jianrong Liu conceived and designed the experiments, authored or reviewed drafts of the article, and approved the final draft.

Chunmei Bai performed the experiments, analyzed the data, prepared figures and/or tables, and approved the final draft.

Yang Li performed the experiments, prepared figures and/or tables, and approved the final draft.

Yanxin Fan conceived and designed the experiments, performed the experiments, analyzed the data, prepared figures and/or tables, authored or reviewed drafts of the article, and approved the final draft.

The following information was supplied relating to ethical approvals (i.e., approving body and any reference numbers):

The Ethics Committee of Fifth Hospital of Shanxi Medical University approved the study (Number: 2023288).

The following information was supplied regarding data availability:

The raw data are available in the Supplemental Files and at NCBI GEO: GSE6798, GSE34526 and GSE137684

The images are available at Figshare: Zhang, xumin (2024). raw figure.zip. figshare. Figure. https://doi.org/10.6084/m9.figshare.26324971.v1.

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
