# Peer review of "Exploring the potential role of ENPP2 in polycystic ovary syndrome and endometrial cancer through bioinformatic analysis"

_PeerJ, doi:10.7717/peerj.18666_

## Round 0.1 · original submission · Major Revisions

Please address all the issues pointed by both reviewers and amend manuscript accordingly.

Reviewer 1 ·

Basic reporting

Peer J #103600

Exploring the Potential Role of ENPP2 in Polycystic Ovary Syndrome and Endometrial Cancer through Bioinformatics Analysis by Zhang et al.
In the current study, the authors find out the ENPP2 gene which is involved in EC and PCOS progression by regulating the hormonal response. For this, they analyzed online available data. They also observed a significant downregulation of ENPP2 expression in the granulosa cells in PCOS compared to the normal patients, while an upregulation of ENPP2 expression was observed in the endometrium of hyperandrogenic PCOS patients relative to the normal. in vitro, WB revealed that 5- 5-dihydrotestosterone (DHT) upregulated ENPP2 expression in Ishikawa (ISK) cells. Additionally, they found that ENPP2 promoted the proliferation, migration, and invasion of ISK cells. Further, IF demonstrated that ENPP2 increased the expression of AR, suggesting a regulatory role for ENPP2 in hormonal response within PCOS and EC.

Overall, the study provides some steps forward in understanding the role of ENPP2 in EC and PCOS. The aim of the study is sound, and the experimental design is reasonable. Experimental data are not sufficient to prove the mechanism of ENPP2 in EC. I have made specific comments below.

1. The author must provide detailed information on the ENNP gene in terms of cancer in the introduction section.
2. Provide detailed information in the method section, especially for cell culture treatment and transfection.
3. Figure quality is not good. Provide the high-resolution figures.
4. Figure legends are not self-explanatory. Therefore, provide a detailed description for each figure legend.

Experimental design

1. Figure 7 (c and d): The author must provide the level of ENNP2 protein in patient tissue samples and compare it with healthy control.
2. Figure 9: The author overexpressed the ENNP2 gene in ISK cells and investigated the proliferation and migration ability of EC cells. The author must downregulate the ENNP2 gene by siRNA and confirm the ENNP2-mediated proliferation and migration ability of EC cells.

Validity of the findings

The author used only one EC cell line in the study. It may be a cell line-specific feature. Therefore, the author must demonstrate the same experiment in other EC cell lines (HEC-1-A, HEC-1-B, or KLE) to validate their findings and prove the role of ENNP2 in EC.

Reviewer 2 ·

Basic reporting

1. Authors provide a full summary of the connection between polycystic ovary syndrome (PCOS) and endometrial carcinoma (EC)and present the reader with a clear hypothesis of a mechanistic link between ferroptosis and the disease progression. Morover, the authors state a clear approach to identify a shared transcriptional signature shared between PCOS and EC, specifically relating to ferroptosis related genes (FRGs) identify new potential clinical biomarkers and therapy targets.
2. Authors claim to have used the publicly avaialbel dataset GSE137648 in this study. However GSE137648 appears to be a DAP-seq dataset of Pseudomonas species. Authors should clarify the correct GSE dataset used for their differential gene expression in PCOS.
3. The English language of following sections should be improved to facilitate comprehension of the manuscript by the PeerJ international community:
a. Line 3: “Bioinformatic Analysis” instead of “Bioinformatics Analysis”
b. Line 20-28: “The shared gene” is misleading here and I suggest the authors refer to ENPP2 directly for clarity
c. Line 263 -264: The grammar and sentence structure makes comprehension difficult.

Experimental design

1. Authors conduct sound statistical analyses to filter all differentially expressed genes (DEGs) to identify the key genes shared between PCOS and EC and provide a visual overview of the workflow of the study in Figure 1.
2. Line 211: Authors identified 741 DEGs while the original source publication of GSE34526 merely identified 211. The authors should comment how they arrived at more than 3-fold higher number of DEGs.
3. Experiments performed and shown in Figure 9c and d are not described in the main text. The use of the ENPP2 inhibitor is described in the methods section but should be described and its reasoning should be included in the results section.
4. Figure 10c: Correlation coefficients whose magnitude are between 0.3 and 0.5 indicate variables which have a low correlation. Therefore R=0.34 does not support the conclusion of “notable correlation” between AR and ENPP2 in the TCGA EC dataset.
5. Figure 10d: The result section does in its current state not adequately describe the experiments and its results in sufficient detail and should be expanded as it clearly shows an increase in AR.
6. Even though the authors state in the text that the AUC for EC for HAMP and HMOX1 were found not be statistically significant the authors should consider including this data in Figure 7 or in the supplement for completeness.
7. Figure 2b: Legend and figure do not indicate the unit of the heatmap scale (red to blue). Authors should also indicate in the legend what type of clustering was performed on the rows of the heatmap. There appears to be fairly extensive heterogeneity between patients of the normal and PCOS groups in both datasets: Some normal samples exhibit a closer expression pattern to PCOS than fellow normal samples and vice versa. The authors should comment in the manuscript text if and to what extent this heterogeneity limits the ability to identify shared differentially expressed genes for further analysis.
8. Figure 3b and Figure 6, Figure 7a: figure y-axis and legend are not indicating the gene expression unit of the plotted values. Legend should also include statistical specifications of the range and whiskers of the box plots and whether mean or median are indicated and if outliers are shown. Statistical significance in * is shown in the figure but legend of significance, statistical test used and associated p values should be included in the legend as well.
9. Line 210: base of log is undefined. Authors should clarify which log was employed for this analysis in the text. Based on Figure 2 it seems it was log2?
10. Figure 2a: y-axis is missing a number scale.

Validity of the findings

1. Given that the authors state they set out to explore association between PCOS and EC and the mechanism of ferroptosis (line 81-84) the authors should either rephrase their motivation to use a filter for ferroptosis in their effort to identify PCOS and EC associated genes or alternatively include experimental results or analysis linking ENPP2 to ferroptosis. I suggest the authors use their cellular model of overexpression of ENPP2 and the ENPP2 inhibitor ONO-8430506 used in this study and evaluate changes in ferroptosis upon for instance carboplatin or cisplatin treatment followed by BODIPY™ 581/591 C11 assay.
2. The authors suggest a pathway where ENPP2 is involved in retinol metabolism and might result in an increased expression or activity of aromatase and thus increased AR expression, which ultimately is responsible for the EC phenotype of increased proliferation, migration, and invasion. I suggest the authors also include aromatase as a readout in the RT-qPCR experiments to support their hypothesis and illuminate this mechanistic link to support this claim.
3. I commend the authors on exploring an additional link to TME modulation in EC and performing extensive analysis to explore their hypothesis that ENPP2 plays a role in modulating tumor associated immune pathways. However, the reported correlations of all parameters does not meet the criteria of strong correlation with correlation coefficients at or below 0.5. Therefore, the language of the conclusion in line 289-290 should be weakened to be in line with the results.
4. Authors achieve a substantial knockdown of AR in their model and present a convincing link between AR signaling and ENPP2 up-regulation at least in vitro.
5. While HAMP and HMOX1 do not validate in the TCGA dataset, the AUC in PCOS is very convincing, the median expression in EC in the TCGA dataset shows an increase of similar extent. Even though this increase is not found to be statistically significant by the authors’s analysis, I would suggest the authors include HAMP and HMOX RT-qPCR experiment in their clinical samples and ISK cell model to explore their potential validity as biomarker despite not meeting the statistical criteria in this specific tested cohort. Identifying these genes as combinatorial biomarkers in addition to ENPP2 might improve the clinical fidelity and strengthen this manuscript. If adding these experiments is beyond the scope of this work, I would suggest the authors discuss this as a limitation of the study in the discussion section and weaken the language singling out ENPP2 as the central diagnostic biomarker gene in PCOS and EC.

Additional comments

1. While ER and PR hormonal therapy is used in eligible EC patients as standard of care (Kokka et al. 2010), AR is not routinely used yet although high rates of AR positivity has been reported (Moatamed et al. 2023). As the authors also suggest in the introduction and discussion, the proposed link of AR signaling and ENPP2 has significant clinical application potential. Upon further studies the usage of ENPP2 as a biomarker to select EC patients for ARSI treatments in future clinical trials could lead to a shift in standard of care treatment.
2. While I recognize the extent of such experiments might be beyond the scope of this study, it would be interesting to see if the effect shown by the results in this study in ISK cell AR knockdown experiments would also be achievable by ARSI treatment (Enzalutamide etc.) followed by RT-qPCR and scratch test/invasion test etc. ? This would independently verify the link between AR and ENPP2 and its impact on proliferation, migration, and invasion.

---

## Round 0.2 · Minor Revisions

Please address the remaining issues pointed by the reviewer #1 and amend manuscript accordingly.

Reviewer 1 ·

Basic reporting

Peer J #103600v2

Exploring the Potential Role of ENPP2 in Polycystic Ovary Syndrome and Endometrial Cancer through Bioinformatics Analysis by Zhang et al.
In this revised version, the author tried to improve the manuscript by providing new data to explain the reviewer’s comments. I am happy that the author provided most of the data to support his findings, as the reviewers asked. Now, it is more convincing. I would like to make some minor changes.
Minor:

Fig 9 (b): Add colony formation and inhibitor treatment detail in the figure legend.

Experimental design

No comment.

Validity of the findings

Fig 9 (b): Provide the bar graph for HEC-1-A cells also like ISK cells.
Fig 9 (c): Also, Compare the OE-ENPP2 group with the OE-ENPP2+ONO-8430506 group in the scratch assay.
Fig 9 (d): Provide the bar graph for HEC-1-A cells also like ISK cells.

Reviewer 2 ·

Basic reporting

The revised manuscript has significantly improved in terms of clarity of language. The authors have meticulously addressed all my original comments and concerns, and have included labels, additional text, figure legends, figure titles, and axes to enhance the manuscript’s readability and understanding.

Experimental design

I am pleased with the author’s detailed responses to my initial comments and concerns, as well as the subsequent revisions made to the manuscript and figures.

Validity of the findings

The authors have successfully addressed my concerns and aligned the manuscript’s text with the results and data. The revised manuscript now provides a clear and accurate representation of the underlying data, with all conclusions fully supported. I believe the revisions have significantly enhanced the paper, making it a valuable contribution to the field.

---

## Round 0.3 · accepted · Accept

All remaining issues were addressed and revised manuscript is acceptable now.